



# Influence of the vapor wall loss on the degradation rate constants in chamber experiments of levoglucosan and other biomass burning markers

Amelie Bertrand[1,2,*], Giulia Stefenelli[3], Simone M. Pieber[3], Emily A. Bruns[3], Brice Temime-Roussel[1],
Jay G. Slowik[3], Henri Wortham[1], André S.H. Prévôt[3], Imad El Haddad[3] and Nicolas Marchand[1]

[1]Aix Marseille Univ, CNRS, LCE, Marseille France
[2]Agence de l'environnement et de la Maîtrise de l'Energie, 20, avenue du Grésillé – BP 90406 49004 Angers cedex 01 France
[3]Laboratory of Atmospheric Chemistry, Paul Scherrer Institute, 5232, Villigen, Switzerland
[*]Now at Laboratory of Atmospheric Chemistry, Paul Scherrer Institute, 5232, Villigen, Switzerland

*Correspondence to*: Nicolas Marchand (nicolas.marchand@univ-amu.fr)

Vapor wall loss has only recently been shown a potentially significant bias in atmospheric chamber studies. Yet, previous works aimed at the determination of the degradation rate of semi-volatile organic compounds (SVOCs) often did not account for this process. Here we evaluate the influence of vapor wall loss on the determination of the gas phase reaction rate $k_{OH}$ of several biomass burning markers (levoglucosan, mannosan, coniferyl aldehyde, 3-guaiacyl propanol, and acetosyringone) with hydroxyl radicals (OH). Emissions from the combustion of beech wood were injected into a 5.5 m³ Teflon atmospheric chamber, and aged for 4 hours (equivalent to 5 – 8 hours in the atmosphere). The particle phase compound concentrations were monitored using a Thermal Desorption Aerosol Gas Chromatograph coupled to a High-Resolution – Time of Flight – Mass Spectrometer (TAG-AMS). The observed depletion of the concentration was later modeled using two different approaches: the previously published approach which does not take into consideration partitioning and vapor wall loss, and an approach with a more complex theoretical framework which integrates all the processes likely influencing the particle phase concentration. We find that with the first approach one fails to predict the measured markers concentration time evolution. With the second approach, we determine that partitioning and vapor wall loss play a predominant role in the particle phase concentration depletion of all the compounds, while the reactivity with OH has a non-significative effect. Furthermore we show that $k_{OH}$ cannot be determined precisely without a strong constraint of the whole set of physical parameters necessary to formally describe the various processes involved. It was found that the knowledge of the saturation mass concentration $C^*$ is especially crucial. Therefore previously published rate constants of levoglucosan and more generally SVOCs with hydroxyl radicals inferred from atmospheric chamber experiments must be, at least, considered with caution.





# 1 Introduction

Biomass burning is known to emit a significant amount of organic aerosol (OA) (Bruns et al., 2015; Sippula, 2010) in the atmosphere with consequences on our health and climate (Kanakidou et al., 2005; Pope and Dockery, 2006). Many efforts have been made to quantify the contribution of biomass burning organic aerosol (BBOA) to ambient OA concentrations. Often, these contributions are estimated using molecular markers, i.e. compounds specific to a source and assumed, at least implicitly, to be stable toward atmospheric oxidation and re-volatilization/partitioning processes. The anhydrosugar levoglucosan is a by-product of the pyrolysis of cellulose and is ubiquitous in our environment. It is a unambiguous organic marker of biomass burning emissions (Simoneit et al., 1999). However, several studies have recently pointed out the apparent lack of stability of the compound towards oxidation by the hydroxyl radical OH. This has been shown in aqueous solution (Hoffmann et al., 2010), on model particles and with particles generated from nebulization in a flow reactor (Kessler et al. 2010, Lai et al. 2014), and with calculations based on quantum chemistry (Bai et al., 2013), as well as its overall lack of stability during aging (Fortenberry et al., 2017; Bertrand et al., 2018). Most pertinent in regards to the work conducted here are the atmospheric chamber experiments performed by Hennigan et al. (2010; 2011). In those, biomass burning emissions were aged under relevant atmospheric conditions in Teflon atmospheric chambers, and the atmospheric lifetime of levoglucosan was estimated to be of 0.7 to 2.2 days. Despite these considerably short lifetimes however, high concentration of levoglucosan are often found in the environment, up to several µg m$^{-3}$ (e.g. Jordan et al., 2006; Puxbaum et al., 2007; Favez et al., 2010; Piot et al., 2012; Crippa et al., 2013; Bonvalot et al., 2016; Bozzetti et al., 2017).

Recent studies demonstrated that vapor losses at the chamber walls can be substantial and can skew our observations towards OA (Matsunaga and Ziemann, 2010; Zhang et al., 2014; Trump et al., 2016; La et al., 2016). The walls of the chamber act as a condensation sink for the condensable material and in essence act as a competing reservoir to the suspended material in the chamber. The extent to which the vapors interact with the walls can cause underestimations as much as a factor of 4 of the secondary organic aerosol (SOA) mass formed (Zhang et al., 2014). In a general manner they influence the concentration of any semi-volatile organic compounds (SVOCs) present in the chamber by causing a depletion of the compound. Vapor wall loss can thus intrinsically modify the chemical composition of the OA measured in an atmospheric chamber.

In the last few years levoglucosan has been re-visited as a SVOC, and authors have attempted to estimate its saturation mass concentration $C^*$(µg m$^{-3}$). $C^*$ is a semi-empirical compound physical property, a key parameter of the partitioning theory (Donahue et al., 2009) which governs the concentration equilibrium of a compound between the gas and the particle phases for a given OA concentration. The saturation mass concentration $C^*$ of SVOCs range between $1 \times 10^{-2}$ and $1 \times 10^2$ µg m$^{-3}$ (Pandis et al., 2013). It is a relatively complex parameter to constrain. To determine the $C^*$ of levoglucosan, May et al. (2012) measured the evaporation of single component particles with a thermodenuder. They determined a $C^*$ of 13 µg m$^{-3}$ at 298 K is consistent with the estimation by the SIMPOL theoretical approach (8 µg m$^{-3}$) (Pankow and Asher, 2008) (at 293



K). In accordance with these results, Ye et al. (2016) investigated the vapor wall loss of levoglucosan in an atmospheric chamber along with other known SVOCs and showed the significant and irreversible loss of the compounds to the walls (on the order of $3.8 \pm 0.3$ h$^{-1}$). Such behavior can possibly explain the very fast degradation rates of levoglucosan calculated by Hennigan et al. (2010) in the absence of vapor wall loss considerations.

In the present paper we investigate further the impact of vapor wall loss on the apparent depletion kinetics of several biomass burning SVOCs, including levoglucosan, mannosan, coniferyl aldehyde, acetosyringone, and 3-guaiacyl propanol. We measured their concentration as a function of OH exposure by means of a Thermal Desorption Aerosol Gas Chromatograph coupled to a High-Resolution – Time of Flight – Mass Spectrometer (TAG – AMS) (Williams et al., 2006; 2014) during atmospheric chamber experiments. In previous publications, we determined the Primary Organic Aerosol

(POA) emission factors and Secondary Aerosol Production Potential (SAPP) and described the overall modification of the molecular fingerprint of BBOA during aging (Bertrand et al., 2017; 2018). Here we model the concentrations of above mentioned SVOCs with and without vapor wall loss/partitioning considerations and compare to our measurements.

## 2 Methods and Materials

Experiments were conducted in the atmospheric chamber of the Paul Scherrer Institute (PSI, Villigen, Switzerland) (Platt et al., 2013; Klein et al., 2016). The full set-up and protocol of our experiments were already described in Bertrand et al. (2017; 2018). Emissions originated from the combustion of beech logs in residential woodstoves. The Modified Combustion Efficiency (MCE) of the combustion varied between 0.83 and 0.95, and was thus considered a mix of flaming and smoldering. The emissions were injected into the atmospheric chamber via heated (140 °C) stainless-steel lines. Prior to

injection, the emissions were diluted by a factor of 10 by an ejector dilutor (DI-1000, Dekati Ltd). The chamber is a 5.5 m³ Teflon bag mounted on an aluminum frame, set to 2 °C (275 K) and with a 50 % relative humidity (RH). A dedicated suite of instruments was deployed for real time or near real time monitoring of particle and gas phase emissions. This included, a TAG-AMS (Aerodyne Research Inc.) for the organic speciation of the organic aerosol, a HR-ToF-AMS (Aerodyne Research Inc.) equipped with a PM$_{2.5}$ aerodynamic inlet lens for the bulk chemical composition of the non-refractory fraction of the

aerosol and operated under standard conditions (i.e. temperature of the vaporizer set at 600 °C, electronic ionization (EI) at 70 eV) with a temporal resolution of 1 minute), an Aethalometer AE33 (Aerosol d.o.o.) (Drinovec et al., 2015) with a time resolution of 1 minute for the black carbon (BC), a Scanning Mobility Particle Sizer (SMPS, CPC 3022, TSI, and custom built DMA) for particle number size distribution information from 16 - 914 nm (with a time resolution of 5 minutes), and a Proton Transfer Reaction – Time of Flight – Mass Spectrometer (PTR-ToF-MS 8000, Ionicon Analytics) operated under

standard conditions (i.e. ion drift pressure at 2.2 mbar and drift field intensity at 125 Td) for the monitoring of the volatile organic compounds (VOCs) (with a time resolution of 1 minute). The Teflon lines sampling the gaseous phase emissions from the atmospheric chamber were temperature controlled at 60 °C to limit condensation losses. After injection, emissions were left static for approximately 30 minutes for homogenization. Nitrous acid (HONO) was then injected continuously in




the chamber at a flow rate of 1 L min$^{-1}$ and photolyzed under a set of $40 \times 100$ W UV lights to initiate the photochemistry by OH radical formation. Emissions were left aging for approximately 4 hours. After each experiment, the atmospheric chamber was set to 100 % RH and flushed overnight ($\approx$ 12 hours) with ozone (1000 ppm) at ambient temperature.

The TAG-AMS (Williams et al., 2006; 2014) enables the on-line collection and analysis of the organic aerosol at the molecular level with a high time resolution. This version of the TAG-AMS also included a system for in-situ derivatization of the most polar compounds (Isaacman et al., 2014). An entire experiment allowed for five to seven measurements by TAG-AMS, one always carried out before photo-oxidation. The sampling time was progressively increased to compensate for the loss of materials to the walls. It ranged between 5 and 25 minutes. The sampling flow rate was set to 2 L min$^{-1}$. An additional line carrying air filtered from a High-Efficiency Particulate Arrestance (HEPA) filter was installed to make up for the missing flow rate. The total sampling flow rate was set to 9 L min$^{-1}$. The sampling line was equipped with a parallel plates charcoal denuder to remove any traces of organic vapor. A series of deuterated standards including adipic acid-D10, phthalic acid-D4, eicosane-D42 and tetracosane-D50 were used for quantification. Authentic standards were injected for positive identification and calibration of the TAG-AMS. Prior to the campaign, tests in the lab allowed us to estimate the uncertainties on the quantification of derivatized compounds at approximately 10 % (based on replicated injection of standards).

Butanol-D9 (1 µL) was added prior to the start of the aging experiment. To account for the dilution by continuous HONO injection, the OH concentration was retrieved based on the differential reactivity of naphthalene ($[C_{10}H_8]H^+$, m/z 129.070) and butanol-D9 ($[C_4D_9]^+$, m/z 66.126), measured by PTR-ToF-MS, and using their respective rate constant with OH ($k_{OH,but} = 3.14 \times 10^{-12}$ cm$^3$ molecule$^{-1}$ s$^{-1}$ and $k_{OH,n} = 2.30 \times 10^{-11}$ cm$^3$ molecule$^{-1}$ s$^{-1}$ (Barmet et al., 2012; Bertrand et al., 2017; 2018). After 4 hours of aging, the integrated OH exposures were in the range of $5 - 8 \times 10^6$ molecule cm$^{-3}$ hours. This is equivalent to 5 - 8 hours of atmospheric aging (on the basis of an average constant OH concentration of $1 \times 10^6$ molecules cm$^{-3}$).

## 3 Results

In aging experiments conducted in atmospheric chambers, SVOCs can undergo different processes, as illustrated in Figure 1 with the example of levoglucosan. The particle phase of the emissions is lost to the walls. The magnitude of the loss is dependent on the rate constant $k_{wall/p}$. According to their saturation mass concentration $C^*$, compounds in the particle phase can also volatilize and react with the hydroxyl radical OH with a rate constant $k_{OH}$. Finally, vapors can also be adsorbed onto the Teflon walls of the chamber with a rate constant $k_{wall/g}$.

Because most of the parameters needed to fully describe the various processes occurring during atmospheric chamber experiments are unknown or subjected to large uncertainties, we model, in a first approach, the evolution of the concentration of levoglucosan in the particle phase as measured by TAG-AMS with only a consideration for the reactivity towards OH and the particle wall loss (Hennigan et al., 2010; 2011; Kessler et al., 2010; Lambe et al., 2010; Weitkamp et al.,



2007). In a second approach we consider all the processes, using a brute-force search approach to determine the unknown parameters.

### 3.1 First approach for levoglucosan without consideration for vapor wall loss

Table 1 reports the conditions of concentration in the chamber for each experiment. The concentration of primary organic aerosol before lights on in the atmospheric chamber ranges from 10 to 122 µg m$^{-3}$. After aging, the total OA mass concentration is increased by a factor of 3.5 to 7, thus a total OA mass concentration ranging between 53 and 495 µg m$^{-3}$. Levoglucosan contributes 14 - 48 % of the POA mass concentration.

The concentrations measured during aging were corrected for particle wall loss following the method developed by Weitkamp et al. (2007) and Hildebrandt et al. (2009). Briefly, the particle loss rate $k_{wall/p}$ is constrained using the decay of an inert particulate tracer, here BC. $k_{wall/p}$ is assumed constant all throughout the experiment and independent from the size of the particles. We determine a rate constant on the order of $2 - 2.5$ hours$^{-1}$ depending on the experiments. This is within the range of values reported by Platt et al. (2013) with the same atmospheric chamber. Assuming the limiting case where vapors only condense on the suspended material, one can estimate a lower bound for the wall loss corrected concentration $C_{i/p\_WLC}$ using:

$$C_{i/p\_WLC}(t) = C_{i,p}(t) + \int_0^t k_{wall/p}(t).C_{i,P}(t).dt \tag{1}$$

where $C_{i/p}$ is the concentration of the particle phase emissions measured by TAG-AMS in µg m$^{-3}$.

Figure 2a shows the particle wall loss corrected (pWLC) concentration of levoglucosan in the particle phase at time *t* normalized to the initial concentration. After an integrated OH exposure of $5 \times 10^6$ molecules cm$^{-3}$ hour, the concentration of levoglucosan had decreased down to $50 - 80$ % of its initial concentration. The loss rate was typically higher within the first hour of aging and the concentration tended toward stabilization from this point onward.

As the concentration of OH stays roughly constant in these experiments ($1 - 2 \times 10^6$ molecules cm$^{-3}$), the reaction of an organic marker with OH in atmospheric chamber experiments is often described as a pseudo-first order reaction (Hennigan et al., 2010; 2011; Kessler et al., 2010; Lambe et al., 2010; Weitkamp et al., 2007). With this approach, the degradation rate corresponds to the slope of the relative decay of the organic marker concentration logarithmically plotted as a function of the OH exposure (Figure 2b). Our data, in regards to the magnitude of the depletion of levoglucosan, are consistent with those of Hennigan et al. (2010; 2011) (at 295 K) with a slope of $2.5 \times 10^{-11}$ cm$^3$ molecules$^{-1}$ s$^{-1}$ which is equivalent to an atmospheric lifetime of 0.5 days (considering an average OH concentration of $1 \times 10^6$ molecules cm$^{-3}$) with lower and upper limit of 0.2 and 1.8 days. In comparison, Hennigan et al. (2010, 2011) determined an atmospheric lifetime for levoglucosan ranging from 0.7 to 2.2 days (Figure 2b).



However, we note the weak correlation between the fit and the experimental data ($R^2 = 0.19$, n = 41, with n the total number of samples). This indicates that a pseudo first order reaction model fails to explain the effective depletion of levoglucosan within the atmospheric chamber during the aging phase. The experiments show a strong depletion within the first two hours of atmospheric aging, but then the concentration remains at a stable level (Exp 2, 3 ,5 and 6). This suggests that this simple approach without considering the whole processes involved cannot fully explain the observed depletion of a compound in the atmospheric chamber.

**3.2 Dynamic approach with consideration for vapor wall loss**

In order to take into account the whole processes occurring in an atmospheric chamber, we developed a more systematic and dynamic approach. The model here aims at predicting the concentration of a marker in the particle phase, in the gas phase, and at the walls, at any time in the atmospheric chamber (from the injection and there on) taking into account the whole processes involved: gas-particle partitioning, particle wall loss, vapor wall loss, and reactivity with the hydroxyl radicals OH.

**3.2.1 Mathematical formalism of the model**

Here, the change in the concentration of a particle phase marker $i$ at steady state conditions is expressed using Equation 2:

$$\frac{dC_{i,p}}{dt} = \left(C_{i,g} - Ceq_{i,g/p}\right).k_{sink} - C_{i,p}.k_{wall/p} \tag{2}$$

where $C_{i,g}$ is the gas phase concentration of a compound $i$ at steady state conditions in µg m⁻³, $Ceq_{i,g/p}$ is the gas phase concentration at equilibrium in µg.m⁻³, and $k_{sink}$ is the condensation sink in s⁻¹. It describes the ability of the suspended particle to remove vapor by condensation and is related to the particle surface area (Erupe et al., 2010; Kulmala et al., 2001) (Equation 3).

$$k_{sink} = 2.\pi.D_{gas}.\sum_n N_n.dp_n.F_n \tag{3}$$

where $D_{gas}$ is the gas phase molecular diffusivity ($10^{-5}$ m² s⁻¹) , $N_n$ is the particle number concentration in m³ in the size class $n$ as measured by the SMPS, $dp_n$ is the particle diameter of the respective size class, and $F_n$ is the Fuchs-Sutugin transitional correction factor. $F_n$ is given by Fuks and Sutugin (1971) (Equation 4).

$$F = \frac{1+Kn}{1+0.3773.Kn+1.33.Kn.\left(\frac{1+Kn}{\alpha}\right)} \tag{4}$$

$K_n$ is the dimensionless Knudsen number derived from Equation 5, and $\alpha$ is the particle mass accommodation coefficient.





$$Kn = \frac{2\lambda}{dp} \tag{5}$$

where $\lambda$ is the gas mean free path (68 nm).

Equation 2 accounts for the gas-particle partitioning and deposition to the wall. On the premise of simplifying the equations we now consider $C_{i,p}$ as the particle wall loss corrected concentration of a compound $i$ in the particle phase (see section 3.1). Equation 2 can therefore be re-written in the following manner:

$$\frac{dC_{i,p}}{dt} = (C_{i,g} - Ceq_{i,g/p}).k_{sink} \tag{6}$$

Gas phase reactivity of organic compounds with OH radicals has been demonstrated to be significantly larger than heterogeneous reactivity (by two or three orders of magnitude higher) (Esteve et al., 2006; Lambe et al., 2009; Hennigan et al., 2011; Socorro et al., 2016). Therefore, in this study, we assume the heterogeneous process to be negligible compared to the gas phase reactions and thus only consider reactions in the gas phase. Taking into account the reactivity of the compound, its partitioning, and the deposition to the wall of the vapors; we can express the change in the concentration of a gas phase marker $C_{i,g}$ at steady state conditions using Equation 7:

$$\frac{dC_{i,g}}{dt} = (Ceq_{i,g/p} - C_{i,g}).k_{sink} + (Ceq_{i,g/w} - C_{i,g}).k_{wall/g} - C_{i,g}.k_{OH}.[OH] \tag{7}$$

where $Ceq_{i,g/w}$ is the gas phase concentration at equilibrium in µg m$^{-3}$ and $k_{wall/g}$ is the vapor wall loss rate in s$^{-1}$. It is assumed constant all throughout the experiment. $1/k_{wall/g}$ is defined as the residence time of the vapors in the atmospheric chamber. $Ceq_{i,g/p}$ and $Ceq_{i,g/w}$ can be formulated using Equations 8 and 9 :

$$Ceq_{i,g/w} = (C_{i,w} + C_{i,g}).(1 - \frac{1}{1 + \frac{C_i^*}{m_{wall}}}) \tag{8}$$

and

$$Ceq_{i,g/p} = (C_{i,p} + C_{i,g}).(1 - \frac{1}{1 + \frac{C_i^*}{C_{OA}}}) \tag{9}$$

where $C_{OA}$ is the particle wall loss corrected organic aerosol concentration in µg m$^{-3}$ measured by the HR-ToF-AMS, $m_{wall}$ is the equivalent organic mass concentration at the wall in µg m$^{-3}$ and $C_{i,w}$ is the concentration of the marker $i$ at the walls in µg m$^{-3}$. The change in the concentration at steady state conditions is expressed using Equation 10:

$$\frac{dC_{i,g/w}}{dt} = (C_{i,g} - Ceq_{i,g/w}).k_{wall/g} \tag{10}$$





The rate constant $k_{OH}$, along with the accommodation coefficient α, the saturation concentration of the marker $C_i^*$, the equivalent organic mass concentration of the wall $m_{wall}$ and the residence time for the vapors $1/k_{wall/g}$ are virtually unknown parameters. Unlike the particle loss rate $k_{wall/p}$ they cannot be easily constrained by experimental measurements. We determine these parameters by a brute-force search. In a brute-force search, successive conditions out of a predefined

range are tested against the observed data in order to determine the optimum conditions. A loop was written in Igor Pro 6.3 (Wave Metrics Inc.) to test for all possible combinations with a set arrangement as shown in Figure 3. While this approach is always likely to yield a solution, it comes with a high computational cost. In order to reduce this computational cost, we initially tested the parameters over a coarse grid. This allowed us to identify the most sensitive parameters. In further iterations, we constrained the range of few parameters on a smaller range and adjusted the resolution of the gridding (Table

10    2).

We use the Root Mean Square Error (RMSE) and mean bias (MB) between predicted and observed value of the particle phase concentration (normalized to the concentration before lights on) to evaluate the performance of the model and determine the best solution. The RMSE is the standard deviation of the residuals (difference between the observed and predicted value) and can be expressed as a percentage using Equation 11:

$$RMSE = \sqrt{\frac{1}{n}\sum_{i=1}^{n}(m-o)^2} \qquad (11)$$

where $n$ is the number of samples (n = 41), $m$ is the predicted value, and $o$ is the observed value. We calculate a general RMSE that accounts for all the samples from every experiment. A well-fitting model should minimize the RMSE. It is here our most important criterion to evaluate the accuracy of the model. The MB evaluates the tendency of the model to overestimate (negative MB) or underestimate (positive MB) the predicted values compared to the measurements.

$$MB = \frac{1}{n}\sum_{i=1}^{n}(m-o) \qquad (12)$$

The upper and lower limits of the range tested for each parameter were defined according to previous contributions made by other groups. The particle mass accommodation coefficient α is generally poorly constrained, although, most authors have typically made use of a particle mass accommodation coefficient α between 0.1 and 1 (Saleh and Khlystov, 2009; May et al., 2012; Ye et al., 2016; Platt et al., 2017). In other works, Julin et al., (2014) determined a coefficient of near

1, and more recently Sinha et al. (2017) estimated a coefficient of 0.1 – 1 for BBOA fresh and aged emissions. In regards to the equivalent organic mass concentration of the wall $m_{wall}$, studies typically use a $m_{wall}$ on the order of a few mg m⁻³, yet Matsunaga and Ziemann (2010) determined significantly higher $m_{wall}$ between 2 and 24 mg m⁻³ (2 mg m⁻³ for alkanes, 10 mg m⁻³ for alcohols, 4 mg m⁻³ for alkenes, and 24 mg m⁻³ for ketones). We broaden their values to include in our testing range 1.6 mg m⁻³ and 25 mg m⁻³ also. The residence time $1/k_{wall/g}$ for the vapors is a function of the relative humidity (RH)

and atmospheric chamber characteristics. Higher RH and active mixing decrease the residence time (Loza et al., 2010).





Authors have determined residence time ranging between several hours and down to a few minutes in the case where the chamber is equipped with an active mixing system (McMurry and Grosjean, 1985; Ye et al. 2016). Ye et al. (2016) determined the residence time could also vary in proportion with the saturation concentration and is therefore compound dependent. Here we initially considered a residence time ranging between 5 and 90 minutes. The work by May et al. (2012) was used as a first assumption to constrain the range of the saturation mass concentration. Considering their value of 13 µg m$^{-3}$ at 298 K and an enthalpy of vaporization $\Delta H_{vap,i}$ of 101 kJ mol$^{-1}$, we calculated a $C_i^*$ of 0.5 µg m$^{-3}$ at 275 K. This constituted the lower limit of the tested range for the $C^*$ of levoglucosan. The upper limit was set at 25 µg m$^{-3}$. Finally, the rate constant $k_{OH}$ was varied between $5 \times 10^{-12}$ and an upper limit of $1 \times 10^{-10}$ cm$^3$ molecule$^{-1}$ sec$^{-1}$ according to the collision theory of reaction rates (Seinfeld and Pandis, 2006).

**3.2.2 Optimization strategy of the parameters for levoglucosan**

3.2.2.1 Coarse Grid – Influence of the parameters

In a first iteration, the parameters are varied on a coarse grid (Table 2). The particle mass accommodation coefficient α is set to either 0.1, 0.5 or 1. The equivalent organic mass concentration at the wall $m_{wall}$ is set to 1.6, 3.2, 6.4, 12.8, 15 or 25 mg m$^{-3}$. The residence time $1/k_{wall/g}$ is set between 5 and 95 minutes with 10 minutes increments. The saturation mass concentration $C_i^*$ is set to either 0.5, 2, 5, 10, 15, 20, or 25 µg m$^{-3}$. Finally, the rate constant $k_{OH}$ is set to either $5 \times 10^{-12}$, $1 \times 10^{-11}$, $3 \times 10^{-11}$, $5 \times 10^{-11}$, $7 \times 10^{-11}$ or $1 \times 10^{-10}$ cm$^3$ molecule$^{-1}$ sec$^{-1}$. Over 8 000 combinations are tested in this iteration.

In this first iteration the RMSE spans 2 orders of magnitude (from 8 % to 351 %, average = 43.2 %) and a MB ranging between -35 % to 286 % (average = 25 %) and greatly depends on the set of parameters used in the model. Therefore, we investigate the mean effect of each parameter on the performance of the model (RMSE) by means of a design of experiments (DOE) analysis in order to narrow down the ranges of the parameters that best fit the experimental data. The analysis was carried out using a full factorial design within the statistical tool Minitab (Minitab 17, Minitab, Inc.). Figure 4 shows the average RMSE obtained for each level of each of the parameters to be optimized. While these plots only display an average response for a given parameter and by no means should be considered as the best optimum parameters, they nonetheless serve to narrow the ranges tested and to get a more general understanding of the importance of the various processes involved.

Overall the model is not sensitive to the particle mass accommodation coefficient $\alpha$ over the range tested. The mean RMSE for each of the three levels, 0.1, 0.5 and 1, are 32.7 %, 34.3 %, and 34.7 % respectively, thus an amplitude between the results of only 2 %. The accommodation coefficient is used to determine the condensation sink $k_{sink}$. The time scale for the condensation sink is on a few seconds to less than a couple of minutes (See Figure S1 in the supplementary information). It increases by approximately a factor of 2 within the range of accommodation coefficient values tested. The residence time $1/k_{wall/g}$ and $C_i^*$ has the highest influence on the response of the model as they contribute to vary the RMSE between 18.4





% and 89.4 % and between 26.6 % and 50 %, thus amplitudes of 71 % and 23 %. Even without considering a residence time of 5 minutes which appears as an extreme, the RMSE still varies with the different levels on an amplitude of 21 %. Finally, the equivalent organic mass concentration of the wall $m_{wall}$ and the rate constant $k_{OH}$ has only a moderate impact within the range tested. The mean RMSE varies on an amplitude of 7 % and 6.5 %.

Typically, within the range tested lower saturation mass concentration between 2 and 10 µg m$^{-3}$ contribute to improve the model performance. At $C_i^* = 0.5$ µg m$^{-3}$, we fail to systematically yield an acceptable result. The model underestimates every time the depletion (MB of 20 % to 30 %). The RMSE varies between 20 % and 35 %. The situation is somewhat more complex in regards to the residence time. A residence time ranging between 10 and 45 minutes increases the performances of the model. Best performances were obtained with a $1/k_{wall/g}$ ranging between 15 and 25 minutes. At $1/k_{wall/g} = 5$

minutes, the model is generally unable to predict the observed data. A look at the effect of the interactions between the parameters (See Figure S1 in the supplementary material) reveals this is especially true with higher saturation mass concentrations $C_i^*$. With a high $C_i^*$, thus assuming the compound is more volatile, and with a high vapor loss rate, the initial depletion is overestimated while the particle phase concentration of the compound later on increases (Figure S2). The residence time does not influence the response of the model in the case of lower saturation mass concentrations (< 5 µg m$^{-3}$)

or as explicitly stated, a compound with a lower volatility have a lower probability to partition in the gas phase, thus its concentration in the particle phase cannot be driven by the vapor loss rate.

3.2.2.2 Fine grid – Results

In a second iteration, the parameters are varied over a finer grid (Table 2). The ranges are selected based upon the observations made after the first iteration. Considering the model is not sensitive to the particle mass accommodation coefficient α, this parameter is set at a constant value of 0.1. The $m_{wall}$ and $k_{OH}$ parameters are left unchanged as no definite conclusion could be drawn from the first iteration. The saturation concentration $C_i^*$ is tested this time on a narrower range, between 1 and 10 µg m$^{-3}$ with an increment of 1 µg m$^{-3}$. The residence time of the vapor is further tested between 10 and 45

minutes. These ranges yield over 3 000 combinations. The RMSE for each is plotted in Figure 5. Overall this finer grid allows to find parameters with better model performances. The RMSE varies between 7.63 % and 32.7 % (average = 19.8 %), and with a MB between -22.2 % and 27.6 % (average = 12.4 %). In this range, the sensitivity of the saturation mass concentration $C_i^*$ and residence time $1/k_{wall/g}$ is lower than on the coarse grid. The response of the model varies respectively on an amplitude of 10 % (17.5 % to 27.5 %) and 14 % (13.5 % to 27.5 %). The influence of the equivalent

organic mass concentration of the wall $m_{wall}$ on the response of the model and the reactivity is decreased as well and is not significant within the studied range (amplitude < 1 % for the $m_{wall}$ and < 3 % for the reactivity).

Based on this iteration, we are able to determine the optimized range of parameters that best fit the experimental data (Table 3) and thus allow us to better understand the mechanism behind the observed depletion of levoglucosan. On Figure 6,



we show the observed and best fit model (RMSE = 7.63 %, MB = 0.8 %, R² = 0.84). Overall, and as in the first iteration only the saturation mass concentration $C_i^*$ and residence time explain the depletion of levoglucosan. Typically, considering a RMSE < 15 %, the optimal $C_i^*$ is between 2 and 10 µg m$^{-3}$ and the $1/k_{wall/g}$ is between 10 and 35 minutes. With a higher degree of confidence (RMSE < 12 %), it is possible to narrow the range of acceptable $C_i^*$ between 3 and 10 µg m$^{-3}$. One has

to consider a RMSE < 10 % to narrow the range of acceptable values for the residence time $1/k_{wall/g}$ to 10 – 25 minutes. The optimized $C_i^*$ range is higher than the values suggested by May et al. (2014) at 275 K, however as stated in section 3.2.1., a saturation concentration of less than 1 µg m$^{-3}$ consistently failed to predict the depletion of levoglucosan observed during the experiment. The optimum range for the residence time is somewhat higher to that observed by Ye et al. (2016) on a chamber of about the same proportion (Teflon, 10 m$^3$, 5.3 min, 273 – 288 K) for levoglucosan but overall constant with the

whole broad of SVOCs tested (15.7 min) (Figure S3). Note, these parameters as evidenced before (Figure S2) are intrinsically linked to one another, and not all combinations within the range proposed will yield satisfactory solutions. For instance in the case of a high $C_i^*$ value, it is only when associated with a high residence time that one might observe a good fit of the data. Overall, these results are more evidences for the semi-volatile nature of levoglucosan and show the depletion of levoglucosan in the chamber can simply be explained by the significant vapor wall loss occurring during the experiment,

rather than the reactivity itself.

While the $m_{wall}$ parameter fail to show a strong influence on the performances of the model at this level, and thus cannot be considered a critic parameter to explain the depletion, we note all solutions with a RMSE < 10 % have a $m_{wall}$ value between 1.6 and 6.4 mg m$^{-3}$, therefore on the lower end of the tested range. Typically, a higher $C_i^*$ associated with a lower $m_{wall}$ does yield a better RMSE. This optimal range is lower than that expected based on the work by Matsunaga and

Ziemann (2010) (10 mg m$^{-3}$ for alcohol, 298 K), but as mentioned before the residence time and saturation concentration considered here implies that a higher $m_{wall}$ would only degrade the performance of the model by a margin of less than 1 %. Therefore, our results do not challenge the conclusions established by Matsunaga and Ziemman (2010).

While $k_{OH}$ has little influence on the overall depletion occurring here, the reactivity rate constant remains an important parameter to determine. Atmospheric implications in the evidence of a high reaction rate of levoglucosan towards OH could

be significant. Determining a meaningful range for the reaction rate constant $k_{OH}$ is however more complex. While here a higher $k_{OH}$ value appeared to overall improve the performances of the model, the RMSE still did not vary by a significant range (< 3 % as mentioned before) when varying the $k_{OH}$ parameter. Furthermore, no trend among the best solutions (RMSE < 10 %) point toward a narrow range of $k_{OH}$ values. To better illustrate the complexity of the matter, a third iteration is ran (ultrafine grid, Table 2). All the parameters but the reaction rate $k_{OH}$ are varied on a grid with only the assumed optimized

range determined in iteration 2. The particle mass accommodation coefficient α is set at 0.1. The saturation mass concentration $C_i^*$ is tested between 3 and 10 µg m$^{-3}$, the equivalent organic mass concentration of the wall $m_{wall}$ is tested between 1.6 – 6.4 mg m$^{-3}$, and the residence time $1/k_{wall/g}$ between 10 – 20 minutes. The reaction rate constant $k_{OH}$ is varied with a finer resolution, between $5 \times 10^{-12}$ and $1 \times 10^{-10}$ cm$^3$ molecules$^{-1}$ sec$^{-1}$ by increment of $5 \times 10^{-12}$ cm$^3$ molecules$^{-1}$




sec$^{-1}$. Over 1 400 combinations are tested in this iteration. The RMSE varies between 7.63 % and 21 % (average = 12.0 %), with a MB ranging from -17.2 % to 16.2 % (average = 0.3 %). While the performances of the model now appear to be optimized with a reaction rate constant ranging between $5 \times 10^{-12}$ and $2 \times 10^{-11}$ cm$^3$ molecules$^{-1}$ sec$^{-1}$, this is important to consider the small amplitude of the mean RMSE for this parameter (less than 1 %). This means that within the tested range,

all the other parameters influence the response of the model more so than the reactivity does. Furthermore, these other parameters also influence the effect of the reactivity on the performances of the model. Here, even a minor change in the conditions impacts the response toward the reactivity, and two sets of conditions relatively similar to one another can generate significant differences in terms of what is a pertinent $k_{OH}$. For instance, Figure 7 shows the RMSE for different levels of the $k_{OH}$ in the case of two sets of conditions where the only parameter changing is the $m_{wall}$ (1.6 to 3.2 mg m$^{-3}$).

With the first set of conditions, the performances of the model are optimized with higher $k_{OH}$ and with a local minima around $7 \times 10^{-12}$ cm$^3$ molecules$^{-1}$ sec$^{-1}$. With the second set of conditions, we obtained a mirror evolution of the RMSE where the performances of the model were optimized with lower rate constant and a local minimum around $3 \times 10^{-12}$ cm$^3$ molecules$^{-1}$ sec$^{-1}$. Note also the range of RMSE at which the solution varied, here, between 10.1 % and 10.9 %, thus an amplitude of less than 1 %. Therefore, not only the reactivity of levoglucosan cannot be considered as the decisive parameter

to explain the depletion of levoglucosan observed here, but we also demonstrate that the rate constant cannot be realistically approached with this method without a better constraint on the vapor wall loss rate and the saturation mass concentration.

### 3.2.3 Extension to other BBOA markers

      The lack of a determining effect by the degradation rate constant $k_{OH}$ on the depletion of the particle phase concentration can be illustrated with other BBOA markers. We tested the model for mannosan and 3 methoxyphenols:

coniferyl aldehyde, acetosyringone, and 3-guaiacyl propanol. The compounds are among the most abundant compounds after levoglucosan detected in the POA (Bertrand et al., 2017). We observed with the TAG-AMS a depletion of these compounds ranging between 40 % and 70 % (Figure S4). To run the model, we assumed the following parameters (Table 2): the particle mass accommodation coefficient α is set to 0.1. The equivalent organic mass concentration at the wall $m_{wall}$ is set to 1.6, 3.2, 6.4, 12.8, 15 or 25 mg m$^{-3}$. The residence time $1/k_{wall/g}$ is set between 5 and 95 minutes with 10 minutes increments.

The saturation mass concentration $C_i^*$ is set to 0.5, 2, 3, 4, 5, 6, 7, 8, 9, 10, 15, 20, or 25 μg m$^{-3}$. Finally, the rate constant $k_{OH}$ is set to either $5 \times 10^{-12}$, $1 \times 10^{-11}$, $3 \times 10^{-11}$, $5 \times 10^{-11}$, $7 \times 10^{-11}$ or $1 \times 10^{-10}$ cm$^3$ molecule$^{-1}$ sec$^{-1}$. A total of 5148 combinations are tested for each compound.

      In Table 4 we report the results of the modelling. The RMSE of the best fit for each compound is reported as the minimum RMSE in the table, and is at under 15 % for the methoxyphenols (respectively 12.4, 11.3, and 8 % for coniferyl

aldehyde, 3-guaiacyl propanol, and acetosyringone) and at 15.4 % for mannosan. Other than the best fit, and as shown on Figure S4 of the supplementary information, we consider that the combinations with a RMSE < 15 % (< 16 % for mannosan) are acceptable solutions as well. They represent less than 13 % of all combinations. We observe that the saturation mass



concentration $C_i^*$ of these sets of combinations range from $3 - 10$ µg m$^{-3}$ for mannosan, $8 - 25$, $4 - 25$ and $2 - 25$ µg m$^{-3}$ for the coniferyl aldehyde, 3-guaiacyl propanol, and acetosyringone. The residence time $1/k_{wall/g}$ ranges from $15 - 25$ minutes for mannosan, 5 - 10, 5 - 15 and $5 - 25$ minutes for the coniferyl aldehyde, 3-guaiacyl propanol, and acetosyringone. Thus similar to our observations made with levoglucosan, we find that only the combinations with a higher saturation mass

concentration $C_i^*$ associated with a lower residence time $1/k_{wall/g}$ can possibly explain the effective depletion of the compounds. It is not possible however to constrain the range of $k_{OH}$. All tested values contain very good solutions. We calculate that on average, a change in the rate constant $k_{OH}$ modifies the performances of the model by less than 0.01 %. Here as well, the rate constant $k_{OH}$ is not a determining parameter to explain the effective concentration depletion.

**4 Conclusions**

In light of the new findings regarding the importance of vapor wall loss in atmospheric chambers (Teflon) and the semi-volatile behavior of many biomass burning markers including levoglucosan, we developed a systematic modelling strategy in order to better understand the depletion of the concentration of these compounds as measured by a TAG-AMS during atmospheric chambers experiments. We attempted to model that depletion taking into account the different processes

involved: vapor wall loss, particle wall loss, partitioning, and reactivity. As many of the parameters are virtually unknown or subjected to high uncertainties we adopted a brute force search approach. This thorough approach allowed us to predict the observed concentration of levoglucosan with a RMSE of 7.63 %, MB of 0.8 % and a R$^2$ = 0.84 between observed and simulated values. We determined a saturation concentration of the levoglucosan in the range of $3 - 10$ µg m$^{-3}$ and a residence time for the vapors on the order of 10 - 15 minutes. The model also succeeded in predicting the evolution of other makers

(RMSE of mannosan = 14.4 %, RMSE of coniferyl aldehyde = 12.4 %, RMSE of 3-guaiacyl propanol = 11.3 % and RMSE of acetosyringone = 8 %. We determined the following $C_i^*$: $3 - 10$ µg m$^{-3}$ for mannosan, $8 - 25$ µg m$^{-3}$, 4 for coniferyl aldehyde, $4 - 25$ µg m$^{-3}$ for 3-guaiacyl propanol, and $2 - 25$ µg m$^{-3}$ for acetosyringone, as well as a residence time $1/k_{wall/g}$ ranging from $15 - 25$ minutes for mannosan, $5 - 10$ minutes for coniferyl aldehyde, $5 - 15$ minutes for 3-guaiacyl propanol and $5 - 25$ minutes for acetosyringone. Overall, this approach clearly demonstrates the predominant role of the partitioning

processes of the compounds towards the gas phase and their subsequent loss at the walls, on both speed and magnitude of the depletion of levoglucosan and that of other markers in the atmospheric chamber. Reactivity towards OH is not, on the other hand, a sensitive parameter and appears to play only a minor role regarding the effective concentration depletion. Thus, the reaction rate $k_{OH}$ cannot be determined precisely without a strong constraint of the whole set of physical parameters necessary to formally describe the various processes involved, and in the first rank of which the saturation concentration $C^*$.

Therefore previously published rate constants of levoglucosan and more generally SVOCs with hydroxyl radicals inferred from atmospheric chamber experiments must be, at least, considered with caution.



**Acknowledgments**

This work was supported by the French Environment and Energy Management Agency (ADEME) project VULCAIN (grant number: 1562C0019). AB also acknowledges ADEME and the Provence-Alpes-Côte d'Azur (PACA) region for their 15 support. PSI acknowledges the financial contribution by the SNF project WOOSHI and the IPR-SHOP SNF starting grant. The authors gratefully acknowledge the MASSALYA instrumental plateform (Aix Marseille Université, lce.univ-amu.fr).

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

30

35





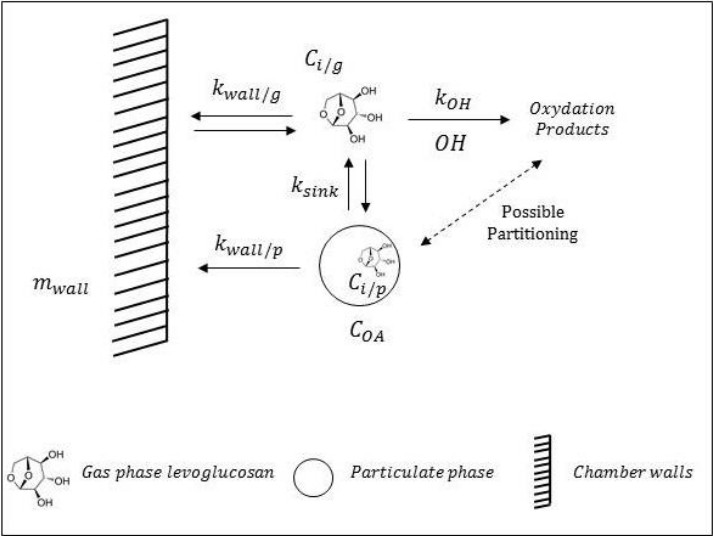

**Figure 1: Theoretical framework representing the interactions between the gas phase, the particle phase, and the walls.**





**Table 1: Organic aerosol concentration before and after aging (corrected for particle wall loss), and levoglucosan concentration measured by TAG-AMS before aging.**

| Exp # | Nb of TAG-AMS samples | $BC_{t=0}$ ($\mu g.m^{-3}$) | $C_{OA,t=0}$ ($\mu g.m^{-3}$) | *$C_{OA,t}$ ($\mu g.m^{-3}$) | OA Enhancement ratio | $C_{levoglucosan,t=0}$ (ng.m$^{-3}$) |
|---|---|---|---|---|---|---|
| Exp 1 | 6 | | 122.3 | 495.4 | 4.1 | 22900 |
| Exp 2 | 8 | | 10.2 | 72.1 | 7.1 | 4900 |
| Exp 3 | 7 | | 40.7 | 143.5 | 3.5 | 5600 |
| Exp 4 | 7 | | 37.7 | 202.1 | 5.4 | 11400 |
| Exp 5 | 6 | | 44.6 | 289.1 | 6.5 | 13900 |
| Exp 6 | 7 | | 9.3 | 53.1 | 5.7 | 3900 |

*values are corrected for the particle wall loss and indicated for an integrated OH exposure of $5.10^6$ molecules cm$^{-3}$ hour



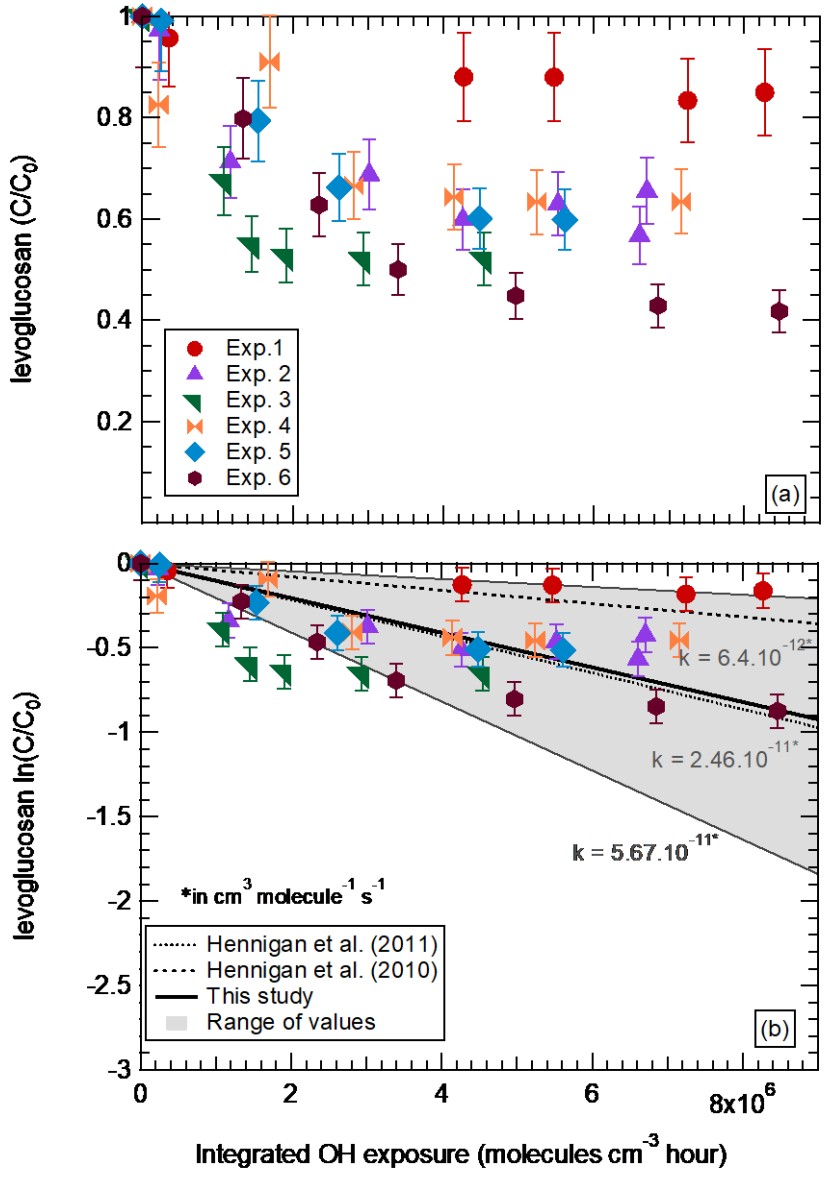

**Figure 2: Particle wall loss corrected (pWLC) concentration of levoglucosan (normalized to its initial concentration) as a function of the integrated OH exposure.**





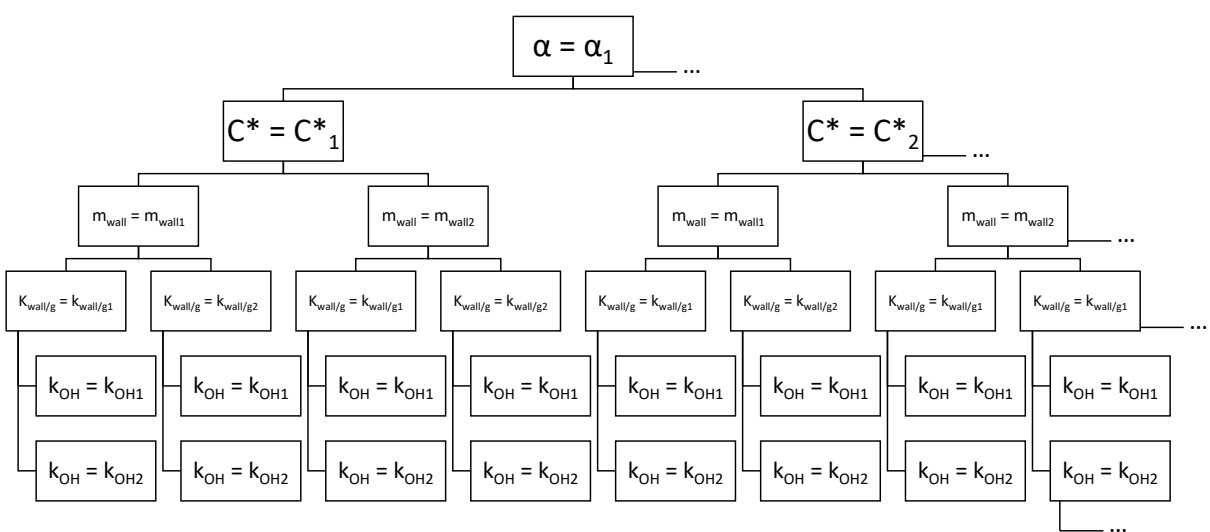

**Figure 3: Illustration of the brute-force search approach applied to solve the model.**



**Table 2: Conditions tested for every iteration of the model in the case of levoglucosan as well as other BBOA markers (mannosan, coniferyl aldehyde, acetosyringone, and 3-guaiacyl propanol).**

| Iteration | Grid | Tested conditions | Nb of combinations |
|---|---|---|---|
| *levoglugosan* | | | |
| 1 | Coarse | $\alpha$ : 0.1, 0.5, 1<br>$C^*$ ($\mu g\ m^{-3}$) : 0.5, 2, 5, 10, 15, 20, 25<br>$m_{wall}$ (mg m$^{-3}$) : 1.6, 3.2, 6.4, 12.8, 15, 25<br>$1/k_{wall/g}$ (min) : 5, 10, 15, 25, 35, 45, 55, 65, 75, 85, 95<br>$k_{OH}$ (cm$^3$ molecule$^{-1}$ sec$^{-1}$) : $5 \times 10^{-12}$, $1 \times 10^{-11}$, $3 \times 10^{-11}$, $5 \times 10^{-11}$, $7 \times 10^{-11}$, $1 \times 10^{-10}$ | 8316 |
| 2 | Fine | $\alpha$ : 0.1<br>$C^*$ ($\mu g\ m^{-3}$) : 1, 2, 3, 4, 5, 6, 7, 8, 9, 10<br>$m_{wall}$ (mg m$^{-3}$) : 1.6, 3.2, 6.4, 12.8, 15, 25<br>$1/k_{wall/g}$ (min) : 5, 10, 15, 20, 25, 30, 35, 40, 45<br>$k_{OH}$ (cm$^3$ molecule$^{-1}$ sec$^{-1}$) : $5 \times 10^{-12}$, $1 \times 10^{-11}$, $3 \times 10^{-11}$, $5 \times 10^{-11}$, $7 \times 10^{-11}$, $1 \times 10^{-10}$ | 2880 |
| 3 | Ultra-Fine | $\alpha$ : 0.1<br>$C^*$ ($\mu g\ m^{-3}$) : 3, 4, 5, 6, 7, 8, 9, 10<br>$m_{wall}$ (mg m$^{-3}$) : 1.6, 3.2, 6.4<br>$1/k_{wall/g}$ (min) : 10, 15, 20<br>$k_{OH}$ (cm$^3$ molecule$^{-1}$ sec$^{-1}$) : $5 \times 10^{-12}$ - $1 \times 10^{-10}$ by increments of $5 \times 10^{-12}$ | 1436 |
| *other BBOA markers* | | | |
| 4 | Fine | $\alpha$ : 0.1<br>$C^*$ ($\mu g\ m^{-3}$) : 0.5, 2, 5, 10, 15, 20, 25<br>$m_{wall}$ (mg m$^{-3}$) : 1.6, 3.2, 6.4, 12.8, 15, 25<br>$1/k_{wall/g}$ (min) : 10, 15, 20<br>$k_{OH}$ (cm$^3$ molecule$^{-1}$ sec$^{-1}$) : $5 \times 10^{-12}$, $1 \times 10^{-11}$, $3 \times 10^{-11}$, $5 \times 10^{-11}$, $7 \times 10^{-11}$, $1 \times 10^{-10}$ | 756 |





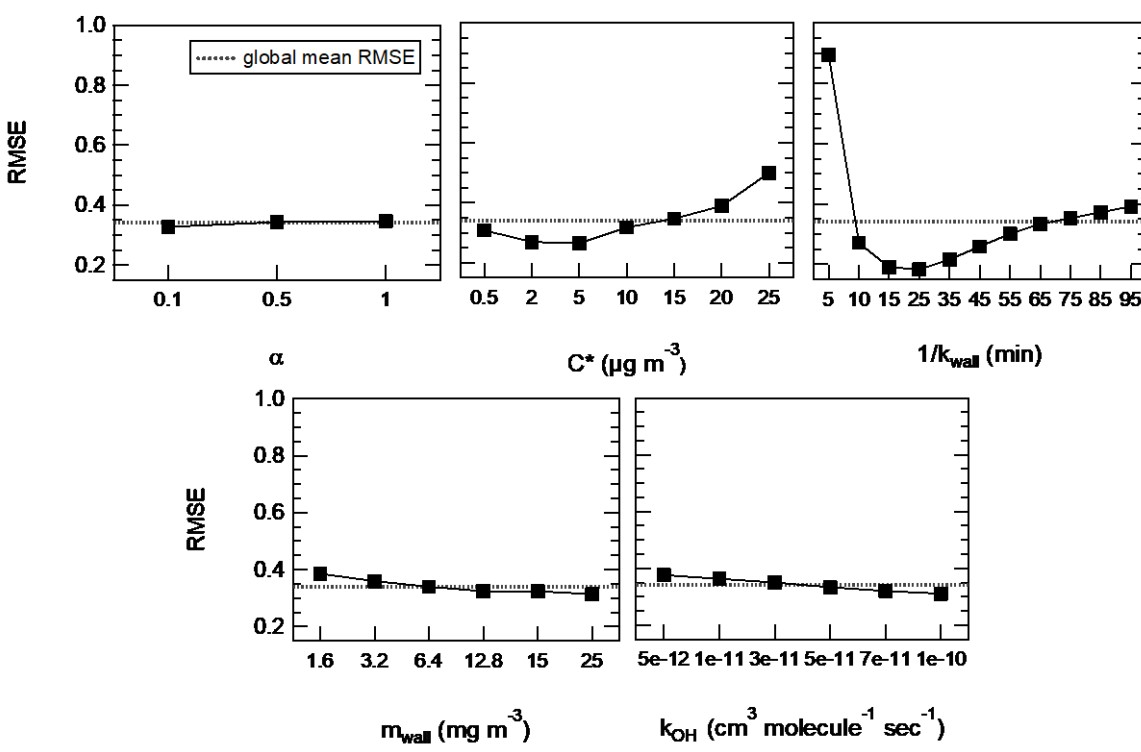

**Figure 4: Influence of the factors on the model in the case of levoglucosan – mean effect plots for RMSE.**





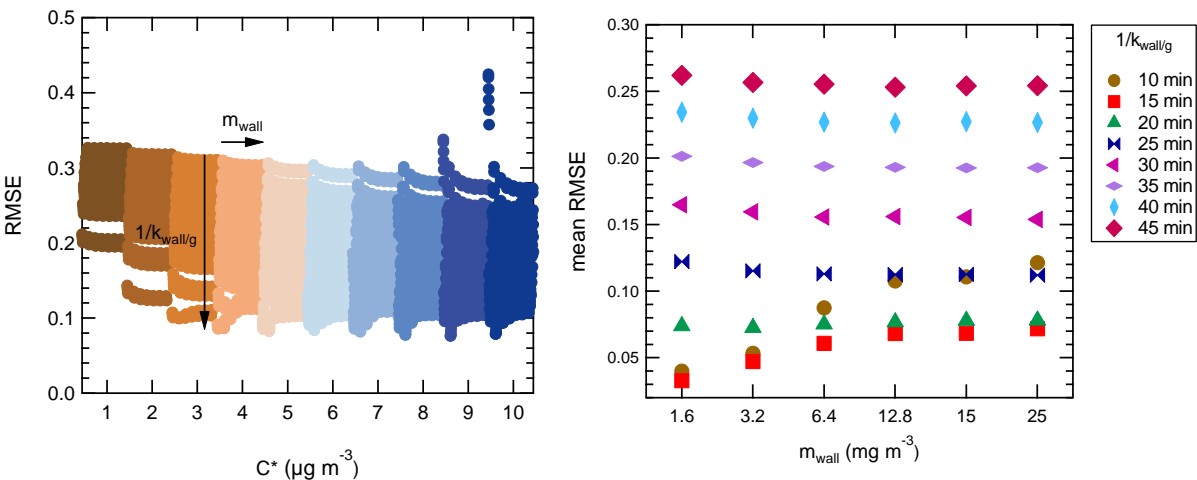

**Figure 5: Influence of the different conditions (tested over a fine grid) on the performances (RMSE) of the model. The accommodation coefficient is set at 0.1. On the left, illustration of the influence of the saturation mass concentration $C_i^*$ parameter. Each condition with a same $C_i^*$ is highlighted a specific color. On the right, illustration of the average influence of the loss rate constant of the vapors $k_{wall/g}$ and equivalent organic mass concentration of the wall $m_{wall}$ on the performances of the model**
10 **(average over the whole range of $C_i^*$ tested).**





**Table 3: Performance of the model for levoglucosan (iteration 2). Initial conditions for this run are presented in Table 2. The accommodation coefficient was set at 0.1. Best fit of the model data with the experimental measurements revealed a RMSE of 7.63 %.**

| Parameter | Response of the model | RMSE < 15 % | RMSE < 12 % | RMSE < 10 % |
|---|---|---|---|---|
| $C*$ ($\mu$g.m$^{-3}$) | sensitive | 2 - 10 | 3 - 10 | 3 - 10 |
| $m_{wall}$ (mg.m$^{-3}$) | not sensitive | 1.6 - 25 | 1.6 - 25 | 1.6 - 6.4 |
| $1/k_{wall/g}$ (min) | sensitive | 10 - 30 | 10 - 25 | 10 - 20 |
| $k_{OH}$ (cm$^3$.molecules$^{-1}$.sec$^{-1}$) | not sensitive | $5 \times 10^{-12}$ - $1 \times 10^{-10}$ | $5 \times 10^{-12}$ - $1 \times 10^{-10}$ | $5 \times 10^{-12}$ - $1 \times 10^{-10}$ |





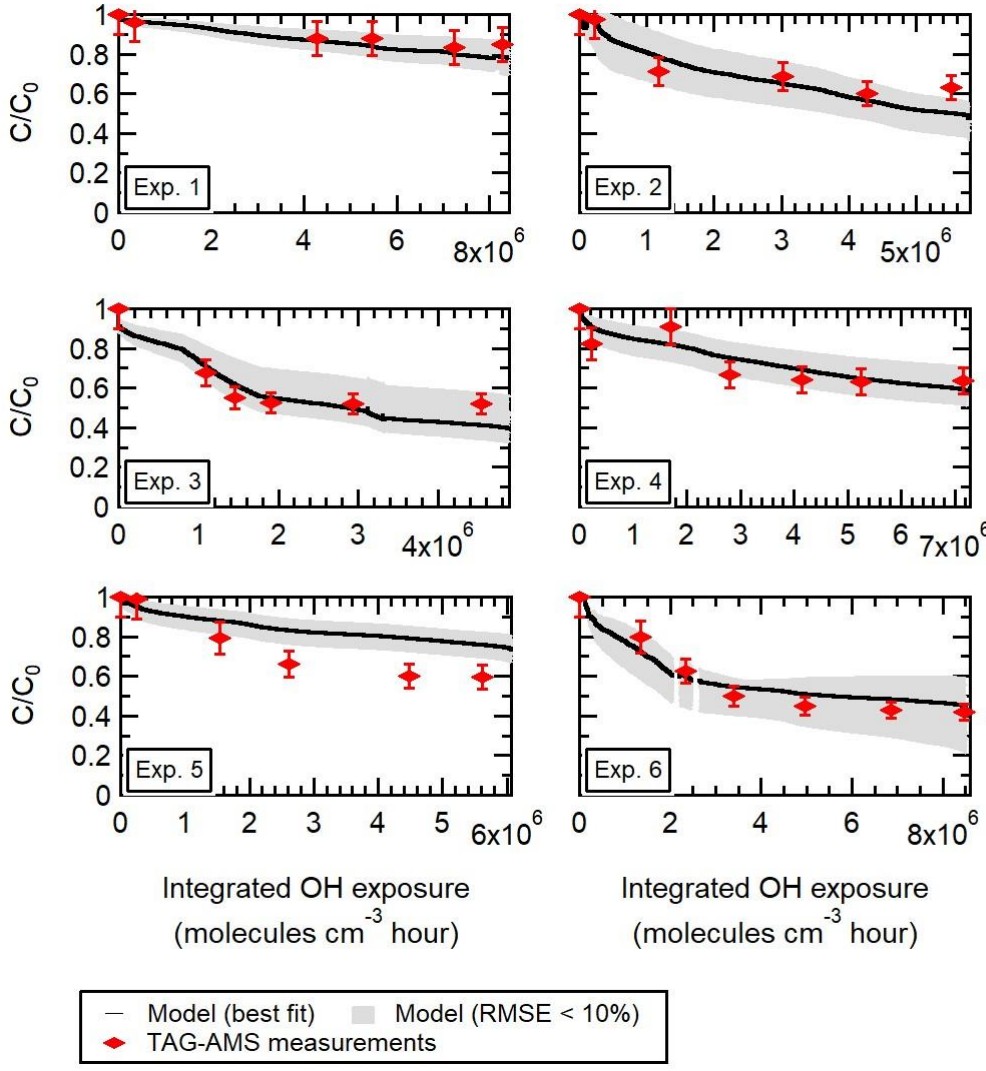

**Figure 6: For each replicate, observed and modeled evolution during aging of the particle phase concentration of levoglucosan**

5    **pWLC (and normalized to the initial concentration). The colored markers are the TAG-AMS measurements. The solid black line**

**represents the best fit (with $\alpha = 0.1$, $C_i^* = 9$ µg.m⁻³, $m_{wall} = 1600$ µg.m⁻³, $1/k_{wall/w} = 15$ min, $k_{OH} = 5 \times 10^{-12}$ cm³ molecules⁻¹ sec⁻¹.**

**RMSE = 7.63 %, mean bias = 0.008). The grey area are all the individual combinations with a RMSE < 10 % (see iteration 2).**





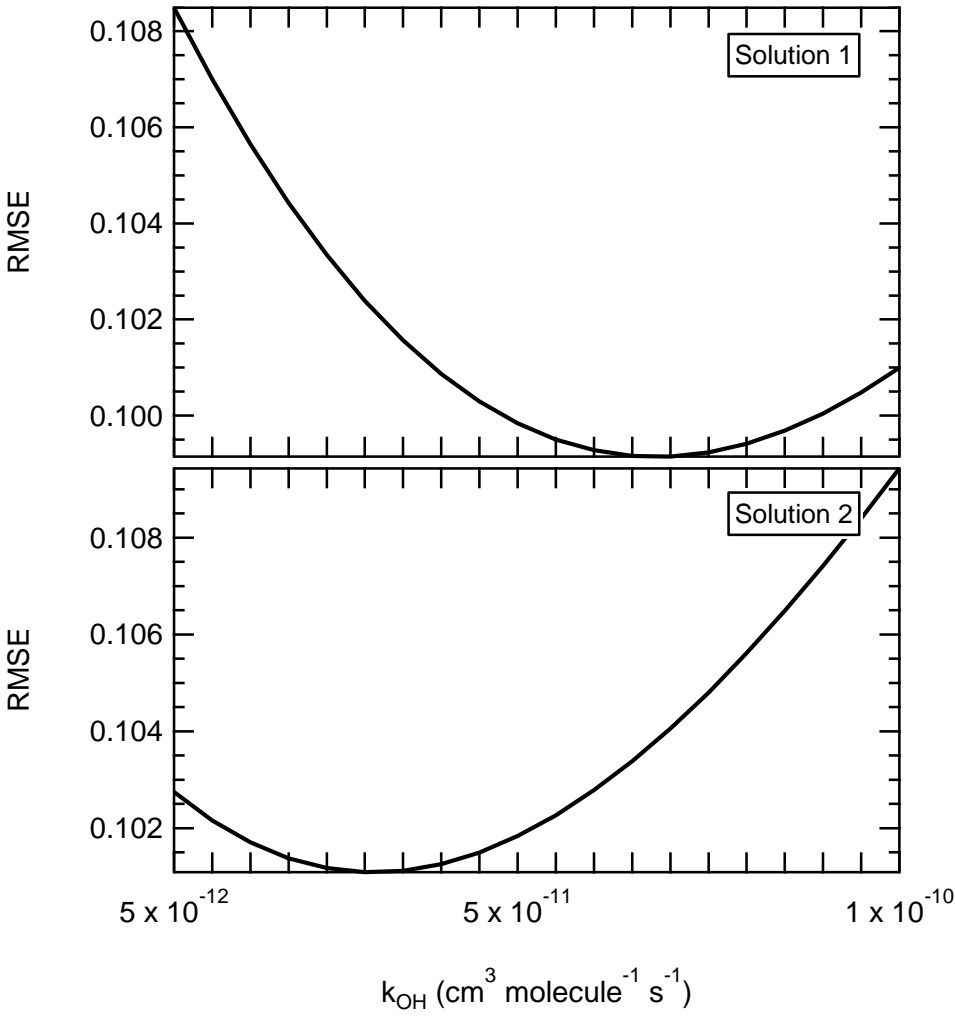

**Figure 7: Effect of the reactivity on the performance of the model. The reaction rate constant $k_{OH}$ was varied between $5 \times 10^{-12}$ and $1 \times 10^{-10}$ cm³ molecules⁻¹ sec⁻¹ by increment of $5 \times 10^{-12}$ cm³ molecules⁻¹ sec⁻¹. Other parameters were set as follow: solution 1 - $\alpha = 0.1$, $C_i^* = 8$ µg m⁻³, $m_{wall} = 1.6$ mg m⁻³, $1/k_{wall/w} = 20$ min. solution 2 - $\alpha = 0.1$, $C_i^* = 8$ µg m⁻³, $m_{wall} = 3.2$ mg m⁻³, $1/k_{wall/w} = 20$ min.**



**Table 4: Performances of the model for BBOA markers (iteration 4). Initial conditions for this run are presented in Table 2. The accommodation coefficient was set at 0.1.**

| Compound | min RMSE (%) | Solutions with a RMSE < 15 %* | | | |
|---|---|---|---|---|---|
| | | C* ($\mu g.m^{-3}$) | $1/k_{wall/g}$ (min) | $m_{wall}$ ($mg.m^{-3}$) | $k_{OH}$ ($cm^3.molecules^{-1}.sec^{-1}$) |
| Mannosan* | 15.4 | 3 - 10 | 15 - 25 | 1.6 - 25 | $5 \times 10^{-12}$ - $1 \times 10^{-10}$ |
| Coniferyl Aldehyde | 12.4 | 8 - 25 | 5 - 10 | 12.8 - 25 | $5 \times 10^{-12}$ - $1 \times 10^{-10}$ |
| 3-Guaiacyl Propanol | 11.3 | 4 - 25 | 5 - 15 | 3.2 - 25 | $5 \times 10^{-12}$ - $1 \times 10^{-10}$ |
| Acetosyringone | 8 | 2 - 25 | 5 - 25 | 1.6 - 25 | $5 \times 10^{-12}$ - $1 \times 10^{-10}$ |

*For Mannosan, RMSE < 16 %

