# Peer review of "Influence of the vapor wall loss on the degradation rate constants in chamber experiments of levoglucosan and other biomass burning markers"

_Atmospheric Chemistry and Physics, 2018_

## Referee Comment (RC1) · Anonymous Referee #1 · 28 Feb 2018

The authors present very valuable data, regarding the influence of vapor wall loss on rate constants in chamber experiments. This is a very important study needed to make sense of laboratory chamber results and make accurate interpretation/comparisons with field studies. The paper addresses relevant scientific questions within the scope of ACP.

However there are some major issues that need to be looked at and proper justification and scientific validity needs to be provided to the assumptions made throughout the paper before publication.

1. The paper talks about laboratory experiments and simulations. However the laboratory studies are not described sufficiently. What are the experimental results? The results section goes to the simulations directly.

2. The particle wall loss rate in these studies is assumed to be constant independent of the size of the particles. The constant used is much higher than what is in literature for wall losses of biomass burning aerosols.

Particle wall loss rate for biomass burning particles for 100 nm size was estimated to range from .147 h-1 to 0.45h-1 See for example chambers KNU: Kyungpook National University (Babar, 2016). TU: Tsinghua University (Shan, 2007). GIG-CAS: Guabgzhon Institute of Geochemistry-Chinese Academy of Science (Wang, 2014). Ilmari University of Eastern Finland (Leskinen, 2015).

For polydisperse aerosol the wall loss rates range $0.17$ h$-1$ , $0.209 \pm 0.018$ h$-1$ , and $0.09$–$0.18$ h$-1$ , respectively (Wang et al., 2014; Paulsen et al., 2005; Cocker et al., 2001). The particle half-life in our remarkably longer than, e.g., the $2.8 \pm 0.8$ h-1 in the PSI mobile chamber (Platt et al., 2013) cited in this work. Most chamber studies have also shown that the loss rate is highly dependent on particle size, and the overall decrease rate of the total number concentration depends on the size distribution of the inspected aerosol, which makes an exact comparison difficult.

Furthermore particles studied in the PSI mobile chamber are diesel emissions which may not be the same as biomass burning aerosols. This distinction should be addressed.

It is not clear how such a very high loss rate affects the simulations, and repeating the experiment using the known values in literature for Biomass burning aerosols may be helpful.

3. A recent work not cited in this paper by Q. Bian , A. A. May , S. M. Kreidenweis , and J. R. Pierce "Investigation of particle and vapor wall-loss effects on controlled woodsmoke smog-chamber experiments "Atmos. Chem. Phys., 15, 11027–11045, 2015. Needs to be considered as this work addresses the same issue and results need to be compared. Furthermore this work uses time and size dependent particle wall loss equations in the simulations.

The size dependent wall loss rate is more realistic and should be used in the simulation and convince readers that the results are independent of the particle size.

The authors also assume vapor wall loss as constant. Again how valid is this assumption? It depends on surface to volume ratio of the chamber and mass accommodation coefficient etc. The authors need to look at the above work by Bian et al. as well and the references provided in there.

4. How can the increase in mass concentration of OA upon aging be explained if there is wall loss?

5. An estimate of the concentration of condensable vapor and its source rate may be important. The assumptions here need to be stated.

6. Conclusions should compare experimental and simulation results in more detail.

Minor points:

Page 2-Line 2: "…… with consequences on our health and climate.." better say with consequences on health and the climate..

Page 2-Line 22: The sentence starting with " the extent to which ….. " is confusing

Page 2-Line 23: "In general manner they influence…. " remove manner

Page 4-Line 26: "The particle phase of the emissions is lost to the walls". It is the particles that are lost not the phase of the emission.. Consider rewriting.

Page 5-line 5: ".. before lights on in…" please change to .. before lights are turned on..

Page 9- line 30 "….. condensation sink is on a few seconds…." Remove "on"

---

## Referee Comment (RC2) · Anonymous Referee #2 · 1 Mar 2018

General Comments

In this manuscript the authors present results of an experimental/modeling study aimed at evaluating the effects of gas-wall partitioning on estimates of gas-phase oxidation rate constants for organic compounds, especially levoglucosan, used as atmospheric markers for biomass burning. The approach was to add biomass burning emissions into a Teflon chamber, expose them to OH radicals generated by HONO photolysis, measure the decay of the marker compounds present in particles, and then simulate the decay using a simple first-order model with corrections for particle wall loss and

then a more complex model that includes various parameters for partitioning of vapors to the particles, particle wall loss, gas-phase reaction with OH, and gas-wall partitioning. The complex model was run many times using values of parameters that fell within a reasonable range based on previous knowledge and the results were then compared to the measured particle-phase concentrations of levoglucosan and some other markers to determine optimum parameter values. The results demonstrate that vapor wall loss is the major mechanism for loss of markers in the chamber and that one cannot accurately determine the gas-phase OH rate constant for loss of markers in the chamber because of its minor effect on decay. These results are important for interpreting results of chamber aging experiments on biomass burning emissions and also field data on biomass burning markers. I think the manuscript is concise and well written, and the technical aspects and interpretations are reasonable. I recommend it be published in ACP after the following minor comments are addressed.

Specific Comments

1. It seems that the model assumes that the chamber is in steady state. Is that a good approximation, and how might it affect the results?

2. Page 9, lines 1–5: There are some more recent references that give useful estimates for timescales for gas-wall partitioning and accommodation coefficients for gas-particle partitioning (Krechmer et al., Env. Sci. Technol., 2016, 2017).

3. Page 11–12: It is probably worth mentioning that calculation of the OH rate constant using the structure-activity relationships of Atkinson and co-workers (e.g. Ziemann and Atkinson, Chem. Soc. Revs., 2012) yields a value at the gas-kinetic limit (>10(–10) cm3 molecule–1 s–1).

4. How do the optimized C* values compare to those calculated using a method such as SIMPOL.1?

Technical Comments

1. Page 6, line 24: "Fuks" should be "Fuchs".

2. Page 13, line 19: "makers" should be "markers".

---

## Author Comment (AC1) · 8 Jun 2018

We thank the Referee for the careful revision and comments which helped to improve the overall quality of the manuscript. A point-by-point answer (in regular typeset) to the referee's remarks (in the *italic typeset*) follows, while changes to the manuscript are indicated in blue font. In the following document, lines references refer to the manuscript version reviewed by the anonymous referee.

**Anonymous Referee #1

*The authors present very valuable data, regarding the influence of vapor wall loss on rate constants in chamber experiments. This is a very important study needed to make sense of laboratory chamber results and make accurate interpretation/comparisons with field studies. The paper addresses relevant scientific questions within the scope of ACP. However there are some major issues that need to be looked at and proper justification and scientific validity needs to be provided to the assumptions made throughout the paper before publication.*

1. *The paper talks about laboratory experiments and simulations. However the laboratory studies are not described sufficiently. What are the experimental results? The results section goes to the simulations directly.*

Results are extensively discussed in two previous papers. In Bertrand et al., (2017)we focused on the influence of the combustion conditions on the emission factors of Primary Organic Aerosol (POA) and the production potential of Secondary Organic Aerosol (SOA). In Bertrand et al., (2018) we characterized with the TAG-AMS the composition at the molecular level of the fresh and aged organic aerosol emissions and its modification all along the aging processes. A third paper is in preparation. Stefenelli et al. (2018, in prep), will present results of our investigation of the contribution to SOA formation of several VOCs precursors.

Following the comment by anonymous reviewer #1, we revised the results section to include in page 4, line 25 of the original manuscript a brief summary of the results of the laboratory measurements as well as the references to the papers mentioned above.

A previous publication already addressed the particulate phase emissions by the different stoves (Bertrand et al., 2017). Briefly, the organic fraction represented 67 – 93 % of the total PM mass observed in the chamber after injection. Black carbon made up for the rest of the composition. The POA concentration in the chamber ranged from 9.3 to 122.3 µg m$^{-3}$ (Table 1). After an aging period corresponding to approximately 5 hours (integrated OH exposure of 5 x 10$^6$ molecule cm-3), we observed an average OA enhancement ratio of 5.3 (3.5 to 7.1). This is equivalent to an OA concentration of 53 - 495 µg m$^{-3}$ after ageing. The TAG-AMS resolved between 26 and 64 % of the total POA mass concentration, but less than 10 % of the total OA mass concentration after aging (integrated OH exposure of 5 x 10$^6$ molecule cm$^{-3}$) (Bertrand et al., 2018). Levoglucosan was the most abundant marker (14 - 42 % of the total POA mass concentration). Its absolute concentration, after particle wall loss correction, decreased significantly over time. We observed a decay of the concentration of levoglucosan by approximately 50 - 80 %. In Bertrand et al., 2018, we report 43 other compounds along with levoglucosan whose concentration decayed during aging. The main compounds include mannosan, coniferyl aldehyde, acetosyringone, and 3-guaiacyl propanol.

Several processes may explain the decay of these SVOCs in an atmospheric chamber. They are detailed in Figure 1. First, particles are lost to the walls and the magnitude of the loss is dependent on the rate constant $k_{wall/p}$. Depending on their saturation mass concentration $C^*$, compounds in the particle phase can also volatilize and react with the hydroxyl radical OH with a rate constant $k_{OH}$. Finally, vapors can also be adsorbed onto the Teflon walls of the chamber with a rate constant $k_{wall/g}$.

Because most of the parameters needed to fully describe the various processes occurring during atmospheric chamber experiments are unknown or subject to large uncertainties, we model, in a first approach, the evolution of the concentration of levoglucosan in the particle phase as measured by TAG-AMS by only considering its reactivity towards OH and the particle wall loss (Hennigan et al., 2010; 2011; Kessler et al., 2010; Lambe et al., 2010; Weitkamp et al., 2007). The aim of this first approach is mostly to compare our own data set with others, previously published (Hennigan et al., 2010; 2011). In a second approach we consider all the processes, using a brute-force search approach to determine the unknown parameters.

*The particle wall loss rate in these studies is assumed to be constant independent of the size of the particles. The constant used is much higher than what is in literature for wall losses of biomass burning aerosols. Particle wall loss rate for biomass burning particles for 100 nm size was estimated to range from .147 h-1 to 0.45h-1 See for example chambers KNU: Kyungpook National University (Babar, 2016). TU: Tsinghua University (Shan, 2007). GIG-CAS: Guabgzhon Institute of Geochemistry-Chinese Academy of Science (Wang, 2014). Ilmari University of Eastern Finland (Leskinen, 2015). For polydisperse aerosol the wall loss rates range 0.17 h−1 , 0.209±0.018 h−1 , and 0.09–0.18 h−1 , respectively (Wang et al., 2014; Paulsen et al., 2005; Cocker et al., 2001).*

*The particle half-life in our remarkably longer than, e.g., the 2.8 ± 0.8 h-1 in the PSI mobile chamber (Platt et al., 2013) cited in this work. Most chamber studies have also shown that the loss rate is highly dependent on particle size, and the overall decrease rate of the total number concentration depends on the size distribution of the inspected aerosol, which makes an exact comparison difficult. Furthermore particles studied in the PSI mobile chamber are diesel emissions which may not be the same as biomass burning aerosols. This distinction should be addressed. It is not clear how such a very high loss rate affects the simulations, and repeating the experiment using the known values in literature for Biomass burning aerosols may be helpful.*

The reviewer is correct. We made a mistake in reporting the particle wall loss rates. Our wall loss rates ranged from $0.2 – 0.3$ h$^{-1}$; the particle half-life is in the range of $2 – 3.5$ hour. therefore, our wall loss rates are in the same range as Platt et al. (2013) and others using this same set-up (Klein et al., 2016 cooking emission and Bruns et al. 2015, biomass burning emission) as well as in the different studies cited by anonymous reviewer #1, although at higher end of the reported range. However we note that with the exception of Babar et al. (2016), the chamber set-up cited by anonymous reviewer #1 are Teflon bags of $28 – 30$ m$^3$. The chamber used here is a 5.5 m$^3$. It is expected that the reduction of the volume and the increase of the surface to volume ratio, would result in higher loss rates.

We apologize for this incorrect reporting and have modified the text accordingly.

We determine a rate constant on the order of $0.2 – 0.3$ hour$^{-1}$ depending on the experiments.

The particle wall loss rates used in this study were inferred from each of the laboratory experiments in which we observed the depletion of levoglucosan. The loss rates were determined based on the observed decay of black carbon. We have included these loss rates in Table 1 of the revised manuscript.

| Exp # | Nb of TAG-AMS samples | $BC_{t = 0}$ (µg.m$^{-3}$) | $C_{OA, t = 0}$ (µg.m$^{-3}$) | *$C_{OA, t}$ (µg.m$^{-3}$) | OA Enhancement ratio | $C_{levoglucosan, t = 0}$ (ng.m$^{-3}$) | $k_{p/wlc}$ (h$^{-1}$) |
|---|---|---|---|---|---|---|---|
| Exp 1 | 6 | 17 | 122 | 495 | 4.1 | 22900 | 0.324 |
| Exp 2 | 8 | 5 | 10 | 72 | 7.1 | 3600 | 0.204 |
| Exp 3 | 7 | 5 | 41 | 143 | 3.5 | 5600 | 0.3 |
| Exp 4 | 7 | 13 | 38 | 202 | 5.4 | 11400 | 0.3 |
| Exp 5 | 6 | 6 | 45 | 289 | 6.5 | 13900 | 0.282 |
| Exp 6 | 7 | 4 | 9 | 53 | 5.7 | 3900 | 0.198 |

*values are corrected for the particulate wall loss and indicated for an integrated OH exposure of $5.10^6$ molecules cm$^{-3}$ hour

Table 1: Organic aerosol concentration before and after aging (corrected for particle wall loss), and levoglucosan concentration measured by TAG-AMS before aging.

2. *A recent work not cited in this paper by Q. Bian , A. A. May , S. M. Kreidenweis , and J. R. Pierce "Investigation of particle and vapor wall-loss effects on controlled wood smoke smog-chamber experiments "Atmos. Chem. Phys., 15, 11027–11045, 2015. Needs to be considered as this work addresses the same issue and results need to be compared.*

We thank anonymous reviewer # 1 for his suggestion. The work by Bian et al. (2015, 2017) does overlap with our studies. We have thus included the following mentions in the revised manuscript:

Page 2, line 20

Recent studies demonstrated that vapor losses at the chamber walls can be substantial, which may lead to false data interpretations and may hinder OA concentration calculations (Matsunaga and Ziemann, 2010; Zhang et al., 2014; Bian et al., 2015; Trump et al., 2016; La et al., 2016)

And page 8, line 24

More recently Sinha et al. (2017) estimated a coefficient of 0.1 – 1 for fresh and aged BBOA emissions while Bian et al. (2015) found a coefficient of 0.01 – 1 were applicable in their own simulation for BBOA emissions.

*Furthermore this work uses time and size dependent particle wall loss equations in the simulations. The size dependent wall loss rate is more realistic and should be used in the simulation and convince readers that the results are independent of the particle size. The authors also assume vapor wall loss as constant. Again how valid is this assumption? It depends on surface to volume ratio of the chamber and mass accommodation coefficient etc. The authors need to look at the above work by Bian et al. as well and the references provided in there.*

We will break our response to this comment in two parts. First we will address the matter of time and size dependency of particle wall loss, second we will focus on the variation of vapor wall loss with time.

**Particle wall losses.** In this work, we have used eBC measured at the end of the experiment to determine a single particles loss rate value per experiment, which was then used to correct the levoglucosan concentrations for wall losses. This correction assumes that eBC is inert, eBC and the organics in the chamber are internally mixed. The particle size distribution shown in figure 1, does suggest that all primary particles are in one mode, which grow with SOA formation. Therefore, there is no indication in our data that BC and OA particles are externally mixed. Evidences from several studies focusing on the mixing state of biomass burning organic aerosol suggest that this is a reasonable assumption (Reid et al., 2005; Schwarz et al., 2008; Raatikainen et al., 2015; Kecorius et al., 2017). This method is commonly used in studies focusing on complex emissions in smog chambers (Grieshop et al., 2009; Hennigan et al., 2010; Platt et al., 2013; Bruns et al., 2015; Klein et al., 2016; Tiitta et al., 2016) or on SOA condensation onto inert seeds such as sulfates (previously injected in a smog

chamber) (Hildebrandt et al, 2009). Therefore, we think that eBC can be used to estimate the wall losses of organic containing particles, under our conditions

[Figure]

**Figure 1. Example of the monomodal distribution of the aerosol (number concentration (top) and mass concentration (bottom) (Experiment 1).**

In table 1, we show that the loss rates between different experiments varies by ~26% (1 geometric standard deviation), most likely due to variation in the particle sizes reached and chamber surface to volume ratio and electrostatic conditions. By considering the loss rate of eBC during different experiments we account for such variability. However, as noted by the reviewer, we assume a constant wall loss rate for each experiment. In Figure 2, we show an example of the exponential fitting of the decay of eBC, from which we retrieve a constant k rate. On the basis that the function well fits the experimental data during the whole aging period, we consider the use of a constant k rate to correct for particle wall loss to be appropriate for these experiments.

[Figure]

**Figure 2. Example of the exponential fit of the decay of BC applied in order to retrieve a constant particle wall loss rate.**

Nevertheless, we consider the comment by anonymous reviewer #2 that the wall loss rate can change over time. To determine the time dependent loss rates, we fit the logarithmic form of the decay on a 30 minutes time interval. We observe that the particle wall loss rate decrease from 0.0052 $min^{-1}$ to 0.0038 $min^{-1}$ over 3.5 hours, on average for all the experiments (inter-experiment variability is removed from this analysis). In Figure 3, we compare both methods of correction: wall loss correction with a constant rate vs. wall loss correction with a time dependent rate. Using a time dependency k rate increased the corrected signal of normalized levoglucosan by < 5 %. Considering the TAG measurement uncertainties (about 10 %) and the different simplifications assumed in the model, we consider that this will not influence the results presented in our study and the main conclusions.

[Figure]

**Figure 3. Wall loss correction of the normalized levoglucosan signal (modeled from experimental data set) using a constant rate (0.0047 min⁻¹) and time dependent rate.**

Following in the comment of anonymous reviewer #1, we have included in the Methods and Material section of the revised manuscript a discussion on our approach to correct the data for particle wall losses. Figure 1, 2 and 3 are also shown in the supplementary information (Figure S1, S2 and S3).

The concentrations measured during aging were corrected for particle wall loss following the method developed by Weitkamp et al. (2007) and Hildebrandt et al. (2009). Briefly, the particle loss rate $k_{wall/p}$ is constrained by fitting with an exponential fit the decay of an inert particulate tracer, here BC7. Here we consider the aerosol to be internally mixed (the black carbon and organic aerosol deposit on the wall at the same rate). The particle size distribution shown in Figure S1, does suggest that all primary particles are in one mode, which grow with SOA formation. Therefore, there is no indication in our data that BC and OA particles are externally mixed. Evidences from several studies focusing on the mixing state of biomass burning organic aerosol suggest that this is a reasonable assumption (Reid et al., 2005; Schwarz et al., 2008; Raatikainen et al., 2015; Kecorius et al., 2017).

The exponential decays and the associated fits are shown in Figure S2 of the supplementary information for each experiment. While a constant $k_{wall/p}$ for each experiment is appropriate to describe the losses of BC, we tested a time dependent $k_{wall/p}$ by fitting the logarithmic form of the decay on a 30 minutes time interval (see Figure S3). Using this time dependency k rate increased the corrected signal of normalized levoglucosan by < 5 %. Considering the TAG measurement uncertainties (about 10 %), we consider that the use of a constant $k_{wall/p}$ for each experiment will not influence the results presented hereafter.

We determine a rate constant on the order of 0.2 – 0.3 hour⁻¹ depending on the experiments (Table 1). This is within the range of values reported by Platt et al. (2013) with the same atmospheric chamber. Assuming the limiting case where vapors only condense on the suspended material, one can estimate a lower bound for the wall loss corrected concentration $C_{i/p\_WLC}$ using:

$$C_{i/p\_WLC}(t) = C_{i,p}(t) + \int_0^t k_{wall/p}(t).C_{i,P}(t).dt$$
(1)

where $C_{i/p}$ is the concentration of the particle phase emissions measured by TAG-AMS in µg m$^{-3}$.

**Vapor Wall Loss**. Indeed the vapor loss rate is dependent on the surface to volume ratio of the chamber (McMurry and Grosjean, 1985; Zhang et al., 2014). This ratio is increased by a factor of 2 during the course of our experiment. Therefore it is expected the vapor wall loss to increase over time.

However, constraining a time dependent vapor wall loss rate is not possible with the size of our dataset and with the type of independent measurements we have. The vapor wall loss rate is one of the five unknown parameter that needs to be constrained. If one were to use a time dependent rate with our current approach, this would mean attempting to optimize a specific vapor wall loss rate for each single data point (ie. 36 different kwall/g), thus leaving us with more unknowns than experimental data available and necessary to constrain the solutions.

While we could consider a more complex model to implement the loss as a function of S/V, we would need the knowledge of the chamber time dependent S/V or the continuous measurement of a non-volatile species in the gas-phase that is readily lost to the walls (e.g. sulfuric acid), which is not really accessible using our instrumentations. As we show, under our conditions we could only provide sufficient constrains mainly to 2 (kwall|g and C*|marker) out of 4 parameters (kwall|g, C|wall, kOH|marker and C*|marker) and enhancing the parameter space would decrease our confidence level about the parameters determined. Therefore, the loss rate of gases determined here should be considered as an average rate for all experiment at different experimental times, which fits with the previously reported vapor wall loss rates. While we are aware that this is a simplification, it nonetheless already shows that the main loss of markers is driven by the walls and not OH, which signifies that, at least under our conditions, vapor wall losses hinders the determination of marker reactivity using chamber experiments. This is the main message of the paper and we are confident that this work remains pertinent as it is, especially in the lights of other work also making use of a constant rate (Zhang et al., 2014b; Ye et al., 2016). Indeed, future work should include more measurements that can better constrain the time dependent vapor wall losses, for precise measurements of markers' reactivity towards OH.

We introduced the following mention in line 16, page 7, to state that we are aware of the limitation of our model on this precise matter.

McMurry and Grosjean, (1985) have defined the vapor wall loss as dependent on the surface to volume ratio (here S/V increased by approximately a factor of 2 during the experiment). Implementing in the model the loss as a function of S/V is however difficult with our instrumentation. Therefore, the vapor wall loss rate $k_{wall/g}$. determined here should be considered as an average rate for all experiments at different experimental times.

3. *How can the increase in mass concentration of OA upon aging be explained if there is wall loss?*

SOA production by biomass burning emission is significant (we determined after aging an enhancement from the primary emissions comprised between 3 and 7 (Bertrand et al, 2017). This is within the same range (1.5 - 6) observed in similar experiments (Grieshop et al., 2009; Heringa et al., 2011; Bruns et al., 2015; Corbin et al., 2015; Tiitta et al., 2016). While thermodynamically the condensation of gases toward the walls is favorable, the condensation of the oxidation products onto the particles is more favorable compared to the walls: condensation sink = [0.02-0.06] s-1 vs. kwall|g = 0.001-0.003 s-1. Therefore, during production the oxidation products will preferentially condense onto the particle phase if they are sufficiently low volatile, but given enough time these

products will leave the particles to the walls. The particles and the walls are only thermodynamically related through the gas phase and the evaporation of low volatility products from the particles to the gas-phase are very slow: for example, considering a compound with a $C^* = 1$ µg m$^{-3}$ at an OA concentration of 100 µg m$^{-3}$, the lifetime of this compound in the chamber towards vapor wall loss is ~15 hours, significantly higher than the experiment duration and the life-time of the particles wall losses. Therefore, a growth can be observed and the evaporation of low volatility products is fairly slow, such that we do not see significant decrease in the aerosol mass after the growth period.

4. *An estimate of the concentration of condensable vapor and its source rate may be important. The assumptions here need to be stated.*

We assume that anonymous reviewer #1 refers to modelling SOA formation requiring the estimation of the production rates of the secondary products and their volatility. However in this work, the decay of levoglucosan is constrained by our measurements of the concentration of levoglucosan in the chamber during aging and the measured OA concentrations, which determines the levoglucosan activity and its evaporation rate. Therefore the present work does not require assumptions about the production rates of condensable gases but uses the measured SOA concentrations as a constrain.

5. *Conclusions should compare experimental and simulation results in more detail.*

We agree with anonymous reviewer #1 that the text needs clarification on how the final simulation compares to the experiments. We have added these elements of the discussion in the results section.

Line 3, Page11 of the manuscript, in addition to what was already stated regarding the statistical performance of the best fit, we wrote:

Considering this best fit only, experiment 1 to 4 were the best represented by the model. The model underestimated the decay of levoglucosan in the case of experiment 5. We note for experiment 1 to 4, the model fails to systematically represent the last data point i.e the model shows a continuous decay of levoglucosan whereas the data points show the concentration is stabilizing.

*Page 2-Line 2: "...... with consequences on our health and climate.." better say with consequences on health and the climate..*

*Page 2-Line 22: The sentence starting with " the extent to which ..... " is confusing*

*Page 2-Line 23: "In general manner they influence.... " remove manner*

*Page 4-Line 26: "The particle phase of the emissions is lost to the walls". It is the particles that are lost not the phase of the emission.. Consider rewriting.*

*Page 5-line 5: ".. before lights on in..." pleasechange to .. before lightsare turned on..*

*Page 9- line 30 "..... condensation sink is on a few seconds...." Remove "on"*

The manuscript was corrected as suggested.

---

## Author Comment (AC2) · 8 Jun 2018

We thank the Referee for the careful revision and comments which helped to improve the overall quality of the manuscript. A point-by-point answer (in regular typeset) to the referee's remarks (in the *italic typeset*) follows, while changes to the manuscript are indicated in blue font. In the following document, lines references refer to the manuscript version reviewed by the anonymous referee.

**Anonymous Referee #2

*General Comments: In this manuscript the authors present results of an experimental/modeling study aimed at evaluating the effects of gas-wall partitioning on estimates of gas-phase oxidation rate constants for organic compounds, especially levoglucosan, used as atmospheric markers for biomass burning. The approach was to add biomass burning emissions into a Teflon chamber, expose them to OH radicals generated by HONO photolysis, measure the decay of the marker compounds present in particles, and then simulate the decay using a simple first-order model with corrections for particle wall loss and then a more complex model that includes various parameters for partitioning of vapors to the particles, particle wall loss, gas-phase reaction with OH, and gas-wall partitioning.*

*The complex model was run many times using values of parameters that fell within a reasonable range based on previous knowledge and the results were then compared to the measured particle-phase concentrations of levoglucosan and some other markers to determine optimum parameter values. The results demonstrate that vapor wall loss is the major mechanism for loss of markers in the chamber and that one cannot accurately determine the gas-phase OH rate constant for loss of markers in the chamber because of its minor effect on decay.*

*These results are important for interpreting results of chamber aging experiments on biomass burning emissions and also field data on biomass burning markers. I think the manuscript is concise and well written, and the technical aspects and interpretations are reasonable. I recommend it be published in ACP after the following minor comments are addressed.*

*Specific Comments*

1. *It seems that the model assumes that the chamber is in steady state. Is that a good approximation, and how might it affect the results?*

The chamber was not operated under steady state conditions, as emissions from the combustion were only injected once and prior to the oxidation. We assume that the anonymous reviewer #2 refers to the section of the manuscripts in which we define several variables as a change of concentration (in the gas phase, particulate phase) at steady state conditions (i.e. line 14 page 6, line 17 page 6). This is a mistake on our part, and these variables should not have been defined in such manner. We removed these descriptions from the manuscript. The modified text reads as follow:

$C_{i,g}$ is the gas phase concentration of a compound $i$ in µg m-3, $Ceq_{i,g/p}$ is the gas phase concentration at equilibrium in µg.m-3

and

Taking into account the reactivity of the compound, its partitioning, and the deposition to the wall of the vapors; we can express the change in the concentration of a gas phase marker $C_{i,g}$ using Equation 7:

and

The change in the concentration is expressed using Equation 10:

2. *Page9, lines1–5: There are some more recent references that give useful estimates for timescales for gas-wall partitioning and accommodation coefficients for gas-particle partitioning (Krechmer et al., Env. Sci. Technol., 2016, 2017).*

These references were added in the manuscript.

In other works, Julin et al., (2014) and Krechmer et al. (2017) determined a coefficient of near 1.

and

Authors have determined residence time comprised between several hours and down to a few minutes in the case where the chamber is equipped with an active mixing system (McMurry and Grosjean, 1985; Ye et al., 2016; Krechmer et al., 2016, 2017).

1. *Page 11–12: It is probably worth mentioning that calculation of the OH rate constant using the structure-activity relationships of Atkinson and co-workers(e.g. Ziemann and Atkinson, Chem. Soc. Revs.,2012) yields a value at the gas-kinetic limit(>10(–10)cm3 molecule–1 s–1).*

We thank anonymous reviewer #2. We have added this clarification in line 9 page 9 of the manuscript.

Finally, the rate constant $k_{OH}$ was varied between $5 \times 10^{-12}$ and an upper limit of $1 \times 10^{-10}$ $cm^3$ molecule$^{-1}$ sec$^{-1}$ according to the collision theory of reaction rates (Seinfeld and Pandis, 2006) although, we note that in their work based on structural-activity relationship, Ziemann and Atkinson (2012) yield a value at the gas-kinetic limit $> 10^{-10}$ $cm^3$ molecule$^{-1}$ sec$^{-1}$.

2. *How do the optimized C* values compare to those calculated using a method such as SIMPOL.1?*

As mentioned line 34, page 2, the SIMPOL model determines a C* of 8 µg m-3 at 298 K for levoglucosan or about 0.5 µg m-3 at 275 K. We determine a C* of 3 µg m-3 at 275 K. Discrepancy between the values yielded by the SIMPOL model and other methods have been commented on before (Kurtén et al., 2016)

*Technical Comments*

1. *Page 6, line 24: "Fuks" should be "Fuchs". 2. Page 13, line 19: "makers" should be "markers".*

Corrected as suggested.

**References**

Julin, J., Winkler, P. M., Donahue, N. M., Wagner, P. E. and Riipinen, I.: Near-Unity Mass Accommodation Coefficient of Organic Molecules of Varying Structure, Environ. Sci. Technol., 48(20), 12083–12089, doi:10.1021/es501816h, 2014.

Kurtén, T., Tiusanen, K., Roldin, P., Rissanen, M., Luy, J.-N., Boy, M., Ehn, M. and Donahue, N.: α-Pinene Autoxidation Products May Not Have Extremely Low Saturation Vapor Pressures Despite High O:C Ratios, J. Phys. Chem. A, 120(16), 2569–2582, doi:10.1021/acs.jpca.6b02196, 2016.

McMurry, P. H. and Grosjean, D.: Gas and aerosol wall losses in Teflon film smog chambers, Environ. Sci. Technol., 19(12), 1176–1182, doi:10.1021/es00142a006, 1985.

Seinfeld, J. H. and Pandis, S. N.: Atmospheric chemistry and physics: from air pollution to climate change, 2nd ed., J. Wiley, Hoboken, N.J., 2006.

Ye, P., Ding, X., Hakala, J., Hofbauer, V., Robinson, E. S. and Donahue, N. M.: Vapor wall loss of semi-volatile organic compounds in a Teflon chamber, Aerosol Sci. Technol., 50(8), 822–834, doi:10.1080/02786826.2016.1195905, 2016.

Ziemann, P. J. and Atkinson, R.: Kinetics, products, and mechanisms of secondary organic aerosol formation, Chem. Soc. Rev., 41(19), 6582, doi:10.1039/c2cs35122f, 2012.